# Graph Random Neural Networks for Semi-Supervised Learning on Graphs

**Wenzheng Feng**[1*], **Jie Zhang**[2*‡], **Yuxiao Dong**[3], **Yu Han**[1], **Huanbo Luan**[1], **Qian Xu**[2],
**Qiang Yang**[2], **Evgeny Kharlamov**[4], **Jie Tang**[1§]

[1] Department of Computer Science and Technology, Tsinghua University
[2]WeBank Co., Ltd    [3]Microsoft Research    [4] Bosch Center for Artificial Intelligence
fwz17@mails.tsinghua.edu.cn, {zhangjie.exe, ericdongyx, yhanthu, luanhuanbo}@gmail.com
{qianxu, qiangyang}@webank.com, evgeny.kharlamov@de.bosch.com, jietang@tsinghua.edu.cn

## Abstract

We study the problem of semi-supervised learning on graphs, for which graph neural networks (GNNs) have been extensively explored. However, most existing GNNs inherently suffer from the limitations of over-smoothing [6, 23, 24, 30], non-robustness [48, 45], and weak-generalization when labeled nodes are scarce. In this paper, we propose a simple yet effective framework—GRAPH RANDOM NEURAL NETWORKS (GRAND)—to address these issues. In GRAND, we first design a random propagation strategy to perform graph data augmentation. Then we leverage consistency regularization to optimize the prediction consistency of unlabeled nodes across different data augmentations. Extensive experiments on graph benchmark datasets suggest that GRAND significantly outperforms state-of-the-art GNN baselines on semi-supervised node classification. Finally, we show that GRAND mitigates the issues of over-smoothing and non-robustness, exhibiting better generalization behavior than existing GNNs. The source code of GRAND is publicly available at `https://github.com/Grand20/grand`.

## 1 Introduction

Graphs serve as a common language for modeling structured and relational data [22], such as social networks, knowledge graphs, and the World Wide Web. Mining and learning graphs can benefit various real-world problems and applications. The focus of this work is on the problem of semi-supervised learning on graphs [46, 20, 10], which aims to predict the categories of unlabeled nodes of a given graph with only a small proportion of labeled nodes. Among its solutions, graph neural networks (GNNs) [20, 17, 35, 1] have recently emerged as powerful approaches. The main idea of GNNs lies in a deterministic feature propagation process to learn expressive node representations.

However, recent studies show that such propagation procedure brings some inherent issues: First, most GNNs suffer from *over-smoothing* [23, 6, 24, 30]. Li et al. show that the graph convolution operation is a special form of Laplacian smoothing [23], and consequently, stacking many GNN layers tends to make nodes' features indistinguishable. In addition, a very recent work [30] suggests that the coupled non-linear transformation in the propagation procedure can further aggravate this issue. Second, GNNs are often *not robust* to graph attacks [48, 45], due to the deterministic propagation adopted in most of them. Naturally, the deterministic propagation makes each node highly dependent with its (multi-hop) neighborhoods, leaving the nodes to be easily misguided by potential data noise and susceptible to adversarial perturbations.

The third issue lies in the general setting of semi-supervised learning, wherein standard training methods (for GNNs) can easily *overfit* the scarce label information [5]. Most efforts to addressing this broad issue are focused on how to fully leverage the large amount of unlabeled data. In computer vision, recent attempts, e.g. MixMatch [3], UDA [40], have been proposed to solve this problem by designing data augmentation methods for consistency regularized training, which have achieved great success in the semi-supervised image classification task. This inspires us to apply this idea into GNNs to facilitate semi-supervised learning on graphs.

In this work, we address these issues by designing graph data augmentation and consistency regularization strategies for semi-supervised learning. Specifically, we present the GRAPH RANDOM NEURAL NETWORKS (GRAND), a simple yet powerful graph-based semi-supervised learning framework.

To effectively augment graph data, we propose random propagation in GRAND, wherein each node's features can be randomly dropped either partially (dropout) or entirely, after which the perturbed feature matrix is propagated over the graph. As a result, each node is enabled to be insensitive to specific neighborhoods, *increasing the robustness of* GRAND. Further, the design of random propagation can naturally separate feature propagation and transformation, which are commonly coupled with each other in most GNNs. This empowers GRAND to safely perform higher-order feature propagation without increasing the complexity, *reducing the risk of over-smoothing for* GRAND. More importantly, random propagation enables each node to randomly pass messages to its neighborhoods. Under the assumption of homophily of graph data [26], we are able to stochastically generate different augmented representations for each node. We then utilize consistency regularization to enforce the prediction model, e.g., a simple Multilayer Perception (MLP), to output similar predictions on different augmentations of the same unlabeled data, *improving* GRAND*'s generalization behavior under the semi-supervised setting.*

Finally, we theoretically illustrate that *random propagation* and *consistency regularization* can enforce the consistency of classification confidence between each node and its multi-hop neighborhoods. Empirically, we also show both strategies can improve the generalization of GRAND, and mitigate the issues of non-robustness and over-smoothing that are commonly faced by existing GNNs. Altogether, extensive experiments demonstrate that GRAND achieves state-of-the-art semi-supervised learning results on GNN benchmark datasets.

## 2 Problem and Related Work

Let $G = (V, E)$ denote a graph, where $V$ is a set of $|V| = n$ nodes and $E \subseteq V \times V$ is a set of $|E|$ edges between nodes. $\mathbf{A} \in \{0, 1\}^{n \times n}$ denotes the adjacency matrix of $G$, with each element $\mathbf{A}_{ij} = 1$ indicating there exists an edge between $v_i$ and $v_j$, otherwise $\mathbf{A}_{ij} = 0$.

**Semi-Supervised Learning on Graphs.** This work focuses on semi-supervised graph learning, in which each node $v_i$ is associated with 1) a feature vector $\mathbf{X}_i \in \mathbf{X} \in \mathbb{R}^{n \times d}$ and 2) a label vector $\mathbf{Y}_i \in \mathbf{Y} \in \{0, 1\}^{n \times C}$ with $C$ representing the number of classes. For semi-supervised classification, $m$ nodes $(0 < m \ll n)$ have observed their labels $\mathbf{Y}^L$ and the labels $\mathbf{Y}^U$ of the remaining $n - m$ nodes are missing. The objective is to learn a predictive function $f : G, \mathbf{X}, \mathbf{Y}^L \rightarrow \mathbf{Y}^U$ to infer the missing labels $\mathbf{Y}^U$ for unlabeled nodes. Traditional approaches to this problem are mostly based on graph Laplacian regularizations [46, 44, 27, 38, 2]. Recently, graph neural networks (GNNs) have emerged as a powerful approach for semi-supervised graph learning, which are reviewed below.

**Graph Neural Networks.** GNNs [15, 33, 20] generalize neural techniques into graph-structured data. The core operation in GNNs is graph propagation, in which information is propagated from each node to its neighborhoods with some deterministic propagation rules. For example, the graph convolutional network (GCN) [20] adopts the propagation rule $\mathbf{H}^{(l+1)} = \sigma(\hat{\mathbf{A}}\mathbf{H}^{(l)}\mathbf{W}^{(l)})$, where $\hat{\mathbf{A}}$ is the symmetric normalized adjacency matrix, $\sigma(.)$ denotes the ReLU function, and $\mathbf{W}^{(l)}$ and $\mathbf{H}^{(l)}$ are the weight matrix and the hidden node representation in the $l^{th}$ layer with $\mathbf{H}^{(0)} = \mathbf{X}$.

The GCN propagation rule could be explained via the approximation of the spectral graph convolutions [4, 18, 8], neural message passing [14], and convolutions on direct neighborhoods [29, 17]. Recent attempts to advance this architecture include GAT [35], GMNN [31], MixHop [1], and GraphNAS [13], etc. In addition, sampling based techniques have also been developed for fast and scalable GNN training, such as GraphSAGE [17], FastGCN [7], AS-GCN [19], and LADIES [47]. The sampling based propagation used in those models may also be used as a graph augmentation

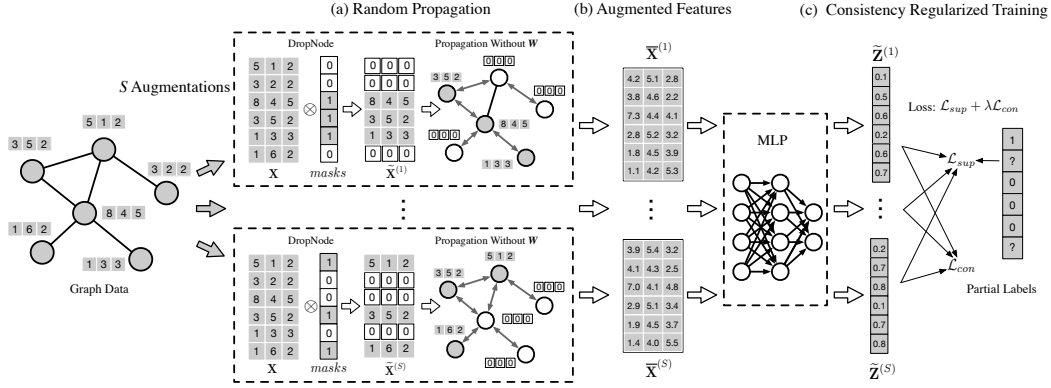

Figure 1: Illustration of GRAND with DropNode as the perturbation method. GRAND designs random propagation (a) to generate multiple graph data augmentations (b), which are further used as consistency regularization (c) for semi-supervised learning.

method. However, its potential effects under semi-supervised setting have not been well-studied, which we try to explore in future work.

**Regularization Methods for GNNs.** Another line of work has aimed to design powerful regularization methods for GNNs, such as VBAT [9], GraphVAT [11], G$^3$NN [25], GraphMix [36], and DropEdge [32]. For example, VBAT [9] and GraphVAT [11] first apply consistency regularized training into GNNs via virtual adversarial training [28], which is highly time-consuming in practice. GraphMix [36] introduces the MixUp strategy [43] for training GNNs. Different from GRAND, GraphMix augments graph data by performing linear interpolation between two samples in the hidden space, and regularizes GNNs by encouraging the model to predict the same interpolation of corresponding labels.

## 3 GRAPH RANDOM NEURAL NETWORKS

We present the GRAPH RANDOM NEURAL NETWORKS (GRAND) for semi-supervised learning on graphs, as illustrated in Figure 1. The idea is to design a propagation strategy (a) to stochastically generate multiple graph data augmentations (b), based on which we present a consistency regularized training (c) for improving the generalization capacity under the semi-supervised setting.

### 3.1 Random Propagation for Graph Data Augmentation

Given an input graph $G$ with its adjacency matrix $\mathbf{A}$ and feature matrix $\mathbf{X}$, the random propagation module generates multiple data augmentations. For each augmentation $\overline{\mathbf{X}}$, it is then fed into the classification model, a two-layer MLP, for predicting node labels. The MLP model can also be replaced with more complex and advanced GNN models, such as GCN and GAT.

**Random Propagation.** There are two steps in random propagation. First, we generate a perturbed feature matrix $\widetilde{\mathbf{X}}$ by randomly dropping out elements in $\mathbf{X}$. Second, we leverage $\widetilde{\mathbf{X}}$ to perform feature propagation for generating the augmented features $\overline{\mathbf{X}}$.

In doing so, each node's features are randomly mixed with signals from its neighbors. Note that the homophily assumption suggests that adjacent nodes tend to have similar features and labels [26]. Thus, the dropped information of a node could be compensated by its neighbors, forming an approximate representation for it in the corresponding augmentation. In other words, random propagation allows us to stochastically generate multiple augmented representations for each node.

In the first step, there are different ways to perturb the input data $\mathbf{X}$. Straightforwardly, we can use the dropout strategy [34], which has been widely used for regularizing neural networks. Specifically, dropout perturbs the feature matrix by randomly setting some elements of $\mathbf{X}$ to 0 during training, i.e., $\widetilde{\mathbf{X}}_{ij} = \frac{\epsilon_{ij}}{1-\delta}\mathbf{X}_{ij}$, where $\epsilon_{ij}$ draws from $Bernoulli(1-\delta)$. In doing so, dropout makes the input feature matrix $\mathbf{X}$ noisy by randomly dropping out its elements without considering graph structures.

To account for the structural effect, we can simply remove some nodes' entire feature vectors—referred to as DropNode, instead of dropping out single feature elements. In other words, DropNode enables each node to only aggregate information from a subset of its (multi-hop) neighbors by completely ignoring some nodes' features, which reduces its dependency on particular neighbors and thus helps increase the model's robustness (Cf. Section 4.5). Empirically, it generates more stochastic data augmentations and achieves better performance than dropout (Cf. Section 4.2).

Formally, in DropNode, we first randomly sample a binary mask $\epsilon_i \sim Bernoulli(1-\delta)$ for each node $v_i$. Second, we obtain the perturbed feature matrix $\widetilde{\mathbf{X}}$ by multiplying each node's feature vector with its corresponding mask, i.e., $\widetilde{\mathbf{X}}_i = \epsilon_i \cdot \mathbf{X}_i$ where $\mathbf{X}_i$ denotes the $i^{th}$ row vector of $\mathbf{X}$. Finally, we scale $\widetilde{\mathbf{X}}$ with the factor of $\frac{1}{1-\delta}$ to guarantee the perturbed feature matrix is in expectation equal to $\mathbf{X}$. Note that the sampling procedure is only performed during training. During inference, we directly set $\widetilde{\mathbf{X}}$ as the original feature matrix $\mathbf{X}$.

In the second step of random propagation, we adopt the mixed-order propagation, i.e., $\overline{\mathbf{X}} = \overline{\mathbf{A}}\widetilde{\mathbf{X}}$, where $\overline{\mathbf{A}} = \sum_{k=0}^{K} \frac{1}{K+1} \hat{\mathbf{A}}^k$ is the average of the power series of $\hat{\mathbf{A}}$ from order 0 to order $K$. This propagation rule enables the model to incorporate more local information, reducing the risk of over-smoothing when compared with directly using $\hat{\mathbf{A}}^K$ [1, 41]. Note that calculating the dense matrix $\overline{\mathbf{A}}$ is computationally inefficient, thus we compute $\overline{\mathbf{X}}$ by iteratively calculating and summing up the product of sparse matrix $\hat{\mathbf{A}}$ and $\hat{\mathbf{A}}^k\widetilde{\mathbf{X}}$ ($0 \le k \le K-1$) in implementation.

With this propagation rule, we could observe that DropNode (dropping the $i^{th}$ row of $\mathbf{X}$) is equivalent to dropping the $i^{th}$ column of $\overline{\mathbf{A}}$. This is similar to DropEdge [32], which aims to address over-smoothing by randomly removing some edges. In practice, DropEdge could also be adopted as the perturbation method here. Specifically, we first generate a corrupted adjacency matrix $\tilde{\mathbf{A}}$ by dropping some elements from $\hat{\mathbf{A}}$, and then use $\tilde{\mathbf{A}}$ to perform mix-order propagation as the substitute of $\hat{\mathbf{A}}$ at each epoch. We empirically compare the effects of different perturbation methods in Section 4.2. By default, we use DropNode as the perturbation method.

**Prediction.** After performing random propagation for $S$ times, we generate $S$ augmented feature matrices $\{\overline{\mathbf{X}}^{(s)}|1 \le s \le S\}$. Each of these augmented data is fed into a two-layer MLP to get the corresponding outputs:

$$\widetilde{\mathbf{Z}}^{(s)} = f_{mlp}(\overline{\mathbf{X}}^{(s)}, \Theta),$$

where $\widetilde{\mathbf{Z}}^{(s)} \in [0,1]^{n \times C}$ denotes the prediction probabilities on $\overline{\mathbf{X}}^{(s)}$ and $\Theta$ are the model parameters.

Observing the data flow from random propagation to the prediction module, it can be realized that GRAND actually separates the feature propagation step, i.e., $\overline{\mathbf{X}} = \overline{\mathbf{A}}\widetilde{\mathbf{X}}$, and transformation step, i.e., $f_{mlp}(\overline{\mathbf{X}}\mathbf{W}, \Theta)$. Note that these two steps are commonly coupled with each other in standard GNNs, that is, $\sigma(\mathbf{A}\mathbf{X}\mathbf{W})$. This separation allows us to perform the high-order feature propagation without conducting non-linear transformations, reducing the risk of over-smoothing (Cf. Section 4.6). A similar idea has been adopted by Klicpera et al. [21], with the difference that they first perform the prediction for each node and then propagate the prediction probabilities over the graph.

### 3.2 Consistency Regularized Training

In graph based semi-supervised learning, the objective is usually to smooth the label information over the graph with regularizations [46, 38, 20], i.e., its loss function is a combination of the supervised loss on the labeled nodes and the graph regularization loss. Given the $S$ data augmentations generated in random propagation, we can naturally design a consistency regularized loss for GRAND's semi-supervised learning.

**Supervised Loss.** With $m$ labeled nodes among $n$ nodes, the supervised objective of the graph node classification task in each epoch is defined as the average cross-entropy loss over $S$ augmentations:

$$\mathcal{L}_{sup} = -\frac{1}{S} \sum_{s=1}^{S} \sum_{i=0}^{m-1} \mathbf{Y}_i^\top \log \widetilde{\mathbf{Z}}_i^{(s)}. \tag{1}$$

**Consistency Regularization Loss.** In the semi-supervised setting, we propose to optimize the prediction consistency among $S$ augmentations for unlabeled data. Considering a simple case of $S =$

---

**Algorithm 1** GRAND

---

**Input:**

   Adjacency matrix $\hat{\mathbf{A}}$, feature matrix $\mathbf{X} \in \mathbb{R}^{n \times d}$, times of augmentations in each epoch $S$, DropNode/dropout probability $\delta$, learning rate $\eta$, an MLP model: $f_{mlp}(\mathbf{X}, \Theta)$.

**Output:**

   Prediction $\mathbf{Z}$.

1: **while** not convergence **do**
2:     **for** $s = 1 : S$ **do**
3:        Pertube the input: $\widetilde{\mathbf{X}}^{(s)} \sim \text{DropNode}(\mathbf{X}, \delta)$.
4:        Perform propagation: $\overline{\mathbf{X}}^{(s)} = \frac{1}{K+1} \sum_{k=0}^{K} \hat{\mathbf{A}}^k \widetilde{\mathbf{X}}^{(s)}$.
5:        Predict class distribution using MLP: $\widetilde{\mathbf{Z}}^{(s)} = f_{mlp}(\overline{\mathbf{X}}^{(s)}, \Theta)$
6:     **end for**
7:     Compute supervised classification loss $\mathcal{L}_{sup}$ via Eq. 1 and consistency regularization loss via Eq. 3.
8:     Update the parameters $\Theta$ by gradients descending: $\Theta = \Theta - \eta \nabla_\Theta (\mathcal{L}_{sup} + \lambda \mathcal{L}_{con})$
9: **end while**
10: Output prediction $\mathbf{Z}$ via: $\mathbf{Z} = f_{mlp}(\frac{1}{K+1} \sum_{k=0}^{K} \hat{\mathbf{A}}^k \mathbf{X}, \Theta)$.

---

2, we can minimize the squared $L_2$ distance between the two outputs, i.e., $\min \sum_{i=0}^{n-1} \|\widetilde{\mathbf{Z}}_i^{(1)} - \widetilde{\mathbf{Z}}_i^{(2)}\|_2^2$. To extend this idea into the multiple-augmentation situation, we first calculate the label distribution center by taking the average of all distributions, i.e., $\overline{\mathbf{Z}}_i = \frac{1}{S} \sum_{s=1}^{S} \widetilde{\mathbf{Z}}_i^{(s)}$. Then we utilize the *sharpening* [3] trick to "guess" the labels based on the average distributions. Specifically, the $i^{th}$ node's guessed probability on the $j^{th}$ class is calculated by:

$$\overline{\mathbf{Z}}_{ij}' = \overline{\mathbf{Z}}_{ij}^{\frac{1}{T}} \bigg/ \sum_{c=0}^{C-1} \overline{\mathbf{Z}}_{ic}^{\frac{1}{T}}, (0 \leq j \leq C-1), \tag{2}$$

where $0 < T \leq 1$ acts as the "temperature" that controls the sharpness of the categorical distribution. As $T \to 0$, the sharpened label distribution will approach a one-hot distribution. We minimize the distance between $\widetilde{\mathbf{Z}}_i$ and $\overline{\mathbf{Z}}_i'$ in GRAND:

$$\mathcal{L}_{con} = \frac{1}{S} \sum_{s=1}^{S} \sum_{i=0}^{n-1} \|\overline{\mathbf{Z}}_i' - \widetilde{\mathbf{Z}}_i^{(s)}\|_2^2. \tag{3}$$

Therefore, by setting $T$ as a small value, we can enforce the model to output low-entropy predictions. This can be viewed as adding an extra entropy minimization regularization into the model, which assumes that the classifier's decision boundary should not pass through high-density regions of the marginal data distribution [16].

**Training and Inference.** In each epoch, we employ both the supervised classification loss in Eq. 1 and the consistency regularization loss in Eq. 3 on $S$ augmentations. The final loss of GRAND is:

$$\mathcal{L} = \mathcal{L}_{sup} + \lambda \mathcal{L}_{con}, \tag{4}$$

where $\lambda$ is a hyper-parameter that controls the balance between the two losses. Algorithm 1 outlines GRAND's training process. During inference, as mentioned in Section 3.1, we directly use the original feature $\mathbf{X}$ for propagation. This is justified because we scale the perturbed feature matrix $\widetilde{\mathbf{X}}$ during training to guarantee its expectation to match $\mathbf{X}$. Hence the inference formula is $\mathbf{Z} = f_{mlp}(\overline{\mathbf{A}}\mathbf{X}, \Theta)$.

**Complexity.** The complexity of random propagation is $\mathcal{O}(Kd(n + |E|))$, where $K$ denotes propagation step, $d$ is the dimension of node feature, $n$ is the number of nodes and $|E|$ denotes edge count. The complexity of its prediction module (two-layer MLP) is $\mathcal{O}(nd_h(d + C))$, where $d_h$ denotes its hidden size and $C$ is the number of classes. By applying consistency regularized training, the total computational complexity of GRAND is $\mathcal{O}(S(Kd(n + |E|) + nd_h(d + C)))$, which is linear with the sum of node and edge counts.

**Limitations.** GRAND is based on the homophily assumption [26], i.e., "birds of a feather flock together", a basic assumption in the literature of graph-based semi-supervised learning [46]. Due to that, however, GRAND may not succeed on graphs with less homophily.

### 3.3 Theoretical Analysis

We theoretically discuss the regularization effects brought by *random propagation* and *consistency regularization* in GRAND. For analytical simplicity, we assume that the MLP used in GRAND has one single output layer, and the task is binary classification. Thus we have $\widetilde{\mathbf{Z}} = \text{sigmoid}(\overline{\mathbf{A}}\widetilde{\mathbf{X}} \cdot \mathbf{W})$, where $\mathbf{W} \in \mathbb{R}^d$ is the learnable parameter vector. For the $i^{th}$ node, the corresponding conditional distribution is $\tilde{z}_i^{y_i}(1 - \tilde{z}_i)^{1-y_i}$, in which $\tilde{z}_i \in \widetilde{\mathbf{Z}}$ and $y_i \in \{0, 1\}$ denotes the corresponding label.

As for the consistency regularization loss, we consider the simple case of generating $S = 2$ augmentations. Then the loss $\mathcal{L}_{con} = \frac{1}{2} \sum_{i=0}^{n-1} \left( \tilde{z}_i^{(1)} - \tilde{z}_i^{(2)} \right)^2$, where $\tilde{z}_i^{(1)}$ and $\tilde{z}_i^{(2)}$ represent the model's two outputs on node $i$ corresponding to the two augmentations, respectively.

With these assumptions, we have the following theorem with proofs in Appendix B.1.

**Theorem 1.** *In expectation, optimizing the unsupervised consistency loss $\mathcal{L}_{con}$ is approximate to optimize a regularization term: $\mathbb{E}_\epsilon(\mathcal{L}_{con}) \approx \mathcal{R}^c(\mathbf{W}) = \sum_{i=0}^{n-1} z_i^2(1-z_i)^2 \text{Var}_\epsilon\left(\overline{\mathbf{A}}_i\widetilde{\mathbf{X}} \cdot \mathbf{W}\right)$.*

**DropNode Regularization.** With DropNode as the perturbation method, we can easily check that $\text{Var}_\epsilon(\overline{\mathbf{A}}_i\widetilde{\mathbf{X}} \cdot \mathbf{W}) = \frac{\delta}{1-\delta}\sum_{j=0}^{n-1}(\mathbf{X}_j \cdot \mathbf{W})^2(\overline{\mathbf{A}}_{ij})^2$, where $\delta$ is drop rate. Then the corresponding regularization term $\mathcal{R}_{DN}^c$ can be expressed as:

$$\mathcal{R}_{DN}^c(\mathbf{W}) = \frac{\delta}{1-\delta} \sum_{j=0}^{n-1} \left[ (\mathbf{X}_j \cdot \mathbf{W})^2 \sum_{i=0}^{n-1} (\overline{\mathbf{A}}_{ij})^2 z_i^2 (1-z_i)^2 \right]. \tag{5}$$

Note that $z_i(1 - z_i)$ (or its square) is an indicator of the classification uncertainty for the $i^{th}$ node, as $z_i(1 - z_i)$ (or its square) reaches its maximum at $z_i = 0.5$ and minimum at $z_i = 0$ or $1$. Thus $\sum_{i=0}^{m-1}(\overline{\mathbf{A}}_{ij})^2 z_i^2(1-z_i)^2$ can be viewed as the weighted average classification uncertainty over the $j^{th}$ node's multi-hop neighborhoods with the weights as the square values of $\overline{\mathbf{A}}$'s elements, which is related to graph structure. On the other hand, $(\mathbf{X}_j \cdot \mathbf{W})^2$—as the square of the input of sigmoid—indicates the classification confidence for the $j^{th}$ node. In optimization, in order for a node to earn a higher classification confidence $(\mathbf{X}_j \cdot \mathbf{W})^2$, it is required that the node's neighborhoods have lower classification uncertainty scores. Hence, *the random propagation with the consistency regularization loss can enforce the consistency of the classification confidence between each node and its multi-hop neighborhoods.*

**Dropout Regularization.** With $\mathbf{X}$ perturbed by dropout, the variance term $\text{Var}_\epsilon(\overline{\mathbf{A}}_i\widetilde{\mathbf{X}} \cdot \mathbf{W}) = \frac{\delta}{1-\delta}\sum_{j=0}^{n-1}\sum_{k=0}^{d-1}\mathbf{X}_{jk}^2\mathbf{W}_k^2(\overline{\mathbf{A}}_{ij})^2$. The corresponding regularization term $\mathcal{R}_{Do}^c$ is

$$\mathcal{R}_{Do}^c(\mathbf{W}) = \frac{\delta}{1-\delta} \sum_{h=0}^{d-1} \mathbf{W}_h^2 \sum_{j=0}^{n-1} \left[ \mathbf{X}_{jh}^2 \sum_{i=0}^{n-1} z_i^2(1-z_i)^2(\overline{\mathbf{A}}_{ij})^2 \right]. \tag{6}$$

Similar to DropNode, this extra regularization term also includes the classification uncertainty $z_i(1-z_i)$ of neighborhoods. However, *we can observe that different from the DropNode regularization, dropout is actually an adaptive $L_2$ regularization for $\mathbf{W}$, where the regularization coefficient is associated with unlabeled data, classification uncertainty, and the graph structure.*

Previous work [37] has also drawn similar conclusions for the case of applying dropout in generalized linear models.

By applying the Cauchy-Schwarz Inequality, we have $\mathcal{R}_{Do}^c \geq \mathcal{R}_{DN}^c$. That is to say, dropout's regularization term is the upper bound of DropNode's. By minimizing this term, dropout can be regarded as an approximation of DropNode.

**Random propagation w.r.t supervised classification loss.** We also discuss the regularization effect of random propagation with respect to the supervised classification loss.

With the previous assumptions, the supervised classification loss is: $\mathcal{L}_{sup} = \sum_{i=0}^{m-1} -y_i \log(\tilde{z}_i) - (1 - y_i)\log(1 - \tilde{z}_i)$. Note that $\mathcal{L}_{sup}$ refers to the perturbed classification loss with DropNode on the node features. By contrast, the original (non-perturbed) classification loss is defined as: $\mathcal{L}_{org} = \sum_{i=0}^{m-1} -y_i \log(z_i) - (1 - y_i)\log(1 - z_i)$, where $z_i = \text{sigmoid}(\overline{\mathbf{A}}_i\mathbf{X} \cdot \mathbf{W})$ is the output with the original feature matrix $\mathbf{X}$. Then we have the following theorem with proof in Appendix B.2.

**Theorem 2.** *In expectation, optimizing the perturbed classification loss $\mathcal{L}_{sup}$ is equivalent to optimize the original loss $\mathcal{L}_{org}$ with an extra regularization term $\mathcal{R}(\mathbf{W})$, which has a quadratic approximation form $\mathcal{R}(\mathbf{W}) \approx \mathcal{R}^q(\mathbf{W}) = \frac{1}{2} \sum_{i=0}^{m-1} z_i(1-z_i)\mathrm{Var}_\epsilon\left(\overline{\mathbf{A}}_i \widetilde{\mathbf{X}} \cdot \mathbf{W}\right).$*

This theorem suggests that DropNode brings an extra regularization loss to the optimization objective. Expanding the variance term, this extra quadratic regularization loss can be expressed as:

$$\mathcal{R}^q_{DN}(\mathbf{W}) = \frac{1}{2}\frac{\delta}{1-\delta}\sum_{j=0}^{n-1}\left[(\mathbf{X}_j \cdot \mathbf{W})^2 \sum_{i=0}^{m-1}(\overline{\mathbf{A}}_{ij})^2\, z_i(1-z_i)\right]. \tag{7}$$

Different from $\mathcal{R}^c_{DN}$ in Eq. 5, the inside summation term in Eq. 7 only incorporates the first $m$ nodes, i.e, the labeled nodes.

## 4 Experiments

### 4.1 Experimental Setup

We follow exactly the same experimental procedure—such as features and data splits—as the standard GNN settings on semi-supervised graph learning [42, 20, 35]. The setup and reproducibility details are covered in Appendix A.

**Datasets.** We conduct experiments on three benchmark graphs [42, 20, 35]—Cora, Citeseer, and Pubmed—and also report results on six publicly available and large datasets in Appendix C.1.

**Baselines.** By default, we use DropNode as the perturbation method in GRAND and compare it with 14 GNN baselines representative of three different categories, as well as its variants:

- Eight graph convolutions: GCN [20], GAT [35], APPNP [21], Graph U-Net [12], SGC [39], MixHop [1], GMNN [31] and GrpahNAS [13].
- Two sampling based GNNs: GraphSAGE [17] and FastGCN [7].
- Four regularization based GNNs: VBAT [9], G$^3$NN [25], GraphMix [36] and Dropedge [32]. We report the results of these methods with GCN as the backbone model.
- Four GRAND variants: GRAND_dropout, GRAND_DropEdge, GRAND_GCN and GRAND_GAT. In GRAND_dropout and GRAND_DropEdge, we use dropout and DropEdge as the perturbation method respectively, instead of DropNode. In GRAND_GCN and GRAND_GAT, we replace MLP with more complex models, i.e., GCN and GAT, respectively.

### 4.2 Overall Results

Table 1 summarizes the prediction accuracies of node classification. Following the community convention [20, 35, 31], the results of baselines are taken from the original works [20, 35, 12, 1, 13, 9, 25, 36, 32, 21]. The results of GRAND are averaged over **100** runs with random weight initializations.

From the top part of Table 1, we can observe that GRAND consistently achieves large-margin outperformance over all baselines across all datasets. Note that the improvements of GRAND over other baselines are all statistically significant (p-value $\ll 0.01$ by a t-test). Specifically, GRAND improves upon GCN by a margin of 3.9%, 5.1%, and 3.7% (absolute differences) on Cora, Citeseer, and Pubmed, while the margins improved by GAT upon GCN were 1.5%, 2.2%, and 0%, respectively. When compared to the very recent regularization based model—DropEdge, the proposed model achieves 2.6%, 3.1%, and 3.1% improvements, while DropEdge's improvements over GCN were only 1.3%, 2.0%, and 0.6%, respectively. To better examine the effectiveness of GRAND in semi-supervised setting, we further evaluate GRAND under different label rates in Appendix C.6.

We observe GRAND_dropout and GRAND_DropEdge also outperform most of baselines, though still lower than GRAND. This indicates DropNode is the best way to generate graph data augmentations in random propagation. Detailed experiments to compare DropNode and dropout under different propagation steps $K$ are shown in Appendix C.4.

We interpret the performance of GRAND_GAT, GRAND_GCN from two perspectives. First, both GRAND_GAT and GRAND_GCN outperform the original GCN and GAT models, demonstrating the

positive effects of the proposed random propagation and consistency regularized training methods. Second, both of them are inferior to GRAND with the simple MLP model, suggesting GCN and GAT are relatively easier to over-smooth than MLP. More analyses can be found in Appendix C.5.

| Method | Cora | Citeseer | Pubmed |
|---|---|---|---|
| GCN [20] | 81.5 | 70.3 | 79.0 |
| GAT [35] | 83.0±0.7 | 72.5±0.7 | 79.0±0.3 |
| APPNP [21] | 83.8±0.3 | 71.6± 0.5 | 79.7 ± 0.3 |
| Graph U-Net [12] | 84.4±0.6 | 73.2±0.5 | 79.6±0.2 |
| SGC [39] | 81.0 ±0.0 | 71.9 ± 0.1 | 78.9 ± 0.0 |
| MixHop [1] | 81.9± 0.4 | 71.4±0.8 | 80.8±0.6 |
| GMNN [31] | 83.7 | 72.9 | 81.8 |
| GraphNAS [13] | 84.2±1.0 | 73.1±0.9 | 79.6±0.4 |
| GraphSAGE [17] | 78.9±0.8 | 67.4±0.7 | 77.8±0.6 |
| FastGCN [7] | 81.4±0.5 | 68.8±0.9 | 77.6±0.5 |
| VBAT [9] | 83.6±0.5 | 74.0±0.6 | 79.9±0.4 |
| $G^3$NN [25] | 82.5±0.2 | 74.4±0.3 | 77.9 ±0.4 |
| GraphMix [36] | 83.9±0.6 | 74.5±0.6 | 81.0±0.6 |
| DropEdge [32] | 82.8 | 72.3 | 79.6 |
| GRAND_dropout | 84.9±0.4 | 75.0±0.3 | 81.7±1.0 |
| GRAND_DropEdge | 84.5±0.3 | 74.4±0.4 | 80.9±0.9 |
| GRAND_GCN | 84.5±0.3 | 74.2±0.3 | 80.0±0.3 |
| GRAND_GAT | 84.3±0.4 | 73.2± 0.4 | 79.2±0.6 |
| GRAND | **85.4±0.4** | **75.4±0.4** | **82.7±0.6** |
| w/o CR | 84.4±0.5 | 73.1±0.6 | 80.9±0.8 |
| w/o mDN | 84.7±0.4 | 74.8±0.4 | 81.0±1.1 |
| w/o sharpening | 84.6±0.4 | 72.2±0.6 | 81.6±0.8 |
| w/o CR & DN | 83.2±0.5 | 70.3±0.6 | 78.5±1.4 |

Table 1: Overall classification accuracy (%).

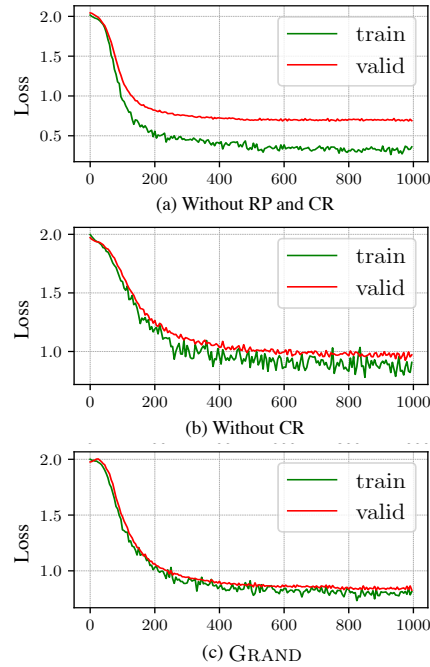

Figure 2: Generalization on Cora ($x$: epoch).

## 4.3 Ablation Study

We conduct an ablation study to examine the contributions of different components in GRAND.

- **Without consistency regularization (CR):** We only use the supervised classification loss, i.e., $\lambda = 0$.
- **Without multiple DropNode (mDN):** Do DropNode once at each epoch, i.e., $S = 1$, meaning that CR only enforces the model to give low-entropy predictions for unlabeled nodes.
- **Without sharpening:** The sharpening trick in Eq. 2 is not used in getting the distribution center, i.e., $T = 1$.
- **Without CR and DropNode (CR & DN):** Remove DropNode (as a result, the CR loss is also removed), i.e., $\delta = 0, \lambda = 0$. In this way, GRAND becomes the combination of deterministic propagation and MLP.

In Table 1, the bottom part summarizes the results of the ablation study, from which we have two observations. First, all GRAND variants with some components removed witness clear performance drops when comparing to the full model, suggesting that each of the designed components contributes to the success of GRAND. Second, GRAND without consistency regularization outperforms almost all eight non-regularization based GCNs and DropEdge in all three datasets, demonstrating the significance of the proposed random propagation technique for semi-supervised graph learning.

## 4.4 Generalization Analysis

We examine how the proposed techniques—random propagation and consistency regularization—improve the model's generalization capacity. To achieve this, we analyze the model's cross-entropy losses on both training and validation sets on Cora. A small gap between the two losses indicates a model with good generalization. Figure 2 reports the results for GRAND and its two variants. We can observe the significant gap between the validation and training losses when without both consistency regularization (CR) and random propagation (RP), indicating an obvious overfitting issue. When applying only the random propagation (without CR), the gap becomes much smaller. Finally, when further adding the CR loss to make it the full GRAND model, the validation loss becomes much closer to the training loss and both of them are also more stable. This observation demonstrates

both the random propagation and consistency regularization can significantly improve GRAND's generalization capability.

## 4.5 Robustness Analysis

We study the robustness of GRAND by generating perturbed graphs with two adversarial attack methods: Random Attack perturbs the graph structure by randomly adding fake edges, and Metattack [49] attacks the graph by removing or adding edges based on meta learning.

Figure 3 presents the classification accuracies of different methods with respect to different perturbation rates on the Cora dataset. We observe that GRAND consistently outperforms GCN and GAT across all perturbation rates on both attacks. When adding 10% new random edges into Cora, we observe only a 7% drop in classification accuracy for GRAND, while 12% for GCN and 37% for GAT. Under Metattack, the gap between GRAND and GCN/GAT also enlarges with the increase of the perturbation rate. This study suggests the robustness advantage of the GRAND model (with or without) consistency regularization over GCN and GAT.

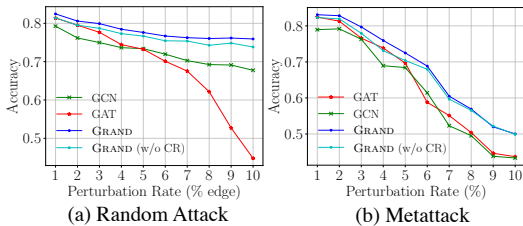
(a) Random Attack  (b) Metattack

Figure 3: Robustness Analysis on Cora.

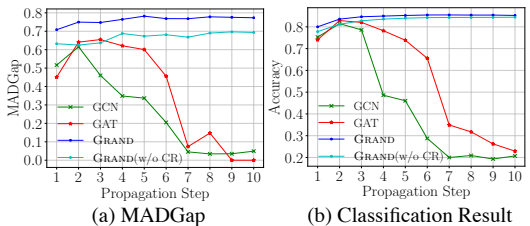
(a) MADGap  (b) Classification Result

Figure 4: Over-Smoothing on Cora

## 4.6 Over-Smoothing Analysis

Many GNNs face the over-smoothing issue—nodes with different labels become indistinguishable—when enlarging the feature propagation steps [23, 6]. We study how vulnerable GRAND is to this issue by using MADGap [6], a measure of the over-smoothness of node representations. A smaller MADGap value indicates the more indistinguishable node representations and thus a more severe over-smoothing issue.

Figure 4 shows both the MADGap values of the last layer's representations and classification results w.r.t. different propagation steps. In GRAND, the propagation step is controlled by the hyperparameter $K$, while for GCN and GAT, it is adjusted by stacking different hidden layers. The plots suggest that as the propagation step increases, both metrics of GCN and GAT decrease dramatically—MADGap drops from ∼0.5 to 0 and accuracy drops from 0.75 to 0.2—due to the over-smoothing issue. However, GRAND behaves completely different, i.e., both the performance and MADGap benefit from more propagation steps. This indicates that GRAND is much more powerful to relieve over-smoothing, when existing representative GNNs are very vulnerable to it.

## 5 Conclusions

In this work, we study the problem of semi-supervised learning on graphs and present the GRAPH RANDOM NEURAL NETWORKS (GRAND). In GRAND, we propose the random propagation strategy to stochastically generate multiple graph data augmentations, based on which we utilize consistency regularization to improve the model's generalization on unlabeled data. We demonstrate its consistent performance superiority over fourteen state-of-the-art GNN baselines on benchmark datasets. In addition, we theoretically illustrate its properties and empirically demonstrate its advantages over conventional GNNs in terms of robustness and resistance to over-smoothing. To conclude, the simple and effective ideas presented in GRAND may generate a different perspective in GNN design, in particular for semi-supervised graph learning. In future work, we aim to further improve the scalability of GRAND with some sampling methods.

## Broader Impact

Over the past years, GNNs have been extensively studied and widely used for semi-supervised graph learning, with the majority of efforts devoted to designing advanced and complex GNN architectures. Instead of heading towards that direction, our work focuses on an alternative perspective by examining whether and how simple and traditional machine learning (ML) techniques can help overcome the common issues that most GNNs faced, including over-smoothing, non-robustness, and weak generalization.

Instead of the nonlinear feature transformations and advanced neural techniques (e.g., attention), the presented GRAND model is built upon dropout (and its simple variant), linear feature propagation, and consistency regularization—the common ML techniques. Its consistent and significant outperformance over 14 state-of-the-art GNN baselines demonstrates the effectiveness of our alternative direction. In addition, our results also echo the recent discovery in SGC [39] to better understand the source of GCNs' expressive power. More importantly, these simple ML techniques in GRAND empower it to be more robust, better avoid over-smoothing, and offer stronger generalization than GNNs.

In light of these advantages, we argue that the ideas in GRAND offer a different perspective in understanding and advancing GNN based semi-supervised learning. For future research in GNNs, in addition to designing complex architectures, we could also invest in simple and traditional graph techniques under the regularization framework which has been widely used in (traditional) semi-supervised learning.

## Footnotes

‡Work performed while at Tsinghua University.

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
