[Supplementary Material]

# Appendix—Graph Random Neural Networks for Semi-Supervised Learning on Graphs

## Contents

## A  Reproducibility

### A.1  Datasets Details

Table 2 summarizes the statistics of the three benchmark datasets — Cora, Citeseer and Pubmed. Our preprocessing scripts for Cora, Citeseer and Pubmed is implemented with reference to the codes of Planetoid [10]. We use exactly the same experimental settings—such as features and data splits—on the three benchmark datasets as literature on semi-supervised graph mining [10, 5, 9] and run 100 trials with 100 random seeds for all results on Cora, Citeseer and Pubmed reported in Section 4. We also evaluate our method on six publicly available and large datasets, the statistics and results are summarized in Appendix C.1.

### A.2  Implementation Details

We make use of PyTorch to implement GRAND and its variants. The random propagation procedure is efficiently implemented with sparse-dense matrix multiplication. The codes of GCN and

Table 2: Benchmark Dataset statistics.

| Dataset | Nodes | Edges | Train/Valid/Test Nodes | Classes | Features |
|---------|-------|-------|------------------------|---------|----------|
| Cora | 2,708 | 5,429 | 140/500/1,000 | 7 | 1,433 |
| Citeseer | 3,327 | 4,732 | 120/500/1,000 | 6 | 3,703 |
| Pubmed | 19,717 | 44,338 | 60/500/1,000 | 3 | 500 |

GRAND_GCN are implemented referring to the PyTorch version of GCN [1]. As for GRAND_GAT and GAT, we adopt the implementation of GAT layer from the PyTorch-Geometric library [2] in our experiments. The weight matrices of classifier are initialized with Glorot normal initializer [2]. We employ Adam [4] to optimize parameters of the proposed methods and adopt early stopping to control the training epochs based on validation loss. Apart from DropNode (or dropout [8]) used in random propagation, we also apply dropout on the input layer and hidden layer of the prediction module used in GRAND as a common practice of preventing overfitting in optimizing neural network. For the experiments on Pubmed, we also use batch normalization [3] to stabilize the training procedure. All the experiments in this paper are conducted on a single NVIDIA GeForce RTX 2080 Ti with 11 GB memory size. Server operating system is Unbuntu 18.04. As for software versions, we use Python 3.7.3, PyTorch 1.2.0, NumPy 1.16.4, SciPy 1.3.0, CUDA 10.0.

### A.3 Hyperparameter Details

**Overall Results in Section 4.2.** GRAND introduces five additional hyperparameters, that is the DropNode probability $\delta$ in random propagation, propagation step $K$, data augmentation times $S$ at each training epoch, sharpening temperature $T$ when calculating consistency regularization loss and the coefficient of consistency regularization loss $\lambda$ trading-off the balance between $\mathcal{L}_{sup}$ and $\mathcal{L}_{con}$. In practice, $\delta$ is always set to 0.5 across all experiments. As for other hyperparameters, we perform hyperparameter search for each dataset. Specifically, we first search $K$ from { 2,4,5,6,8}. With the best selection of $K$, we then search $S$ from {2,3,4}. Finally, we fix $K$ and $S$ to the best values and take a grid search for $T$ and $\lambda$ from {0.1, 0.2, 0.3,0.5} and {0.5, 0.7, 1.0} respectively. For each search of hyperparameter configuration, we run the experiments with 20 random seeds and select the best configuration of hyperparameters based on average accuracy on validation set. Other hyperparameters used in our experiments includes learning rate of Adam, early stopping patience, L2 weight decay rate, hidden layer size, dropout rates of input layer and hidden layer. We didn't spend much effort to tune these hyperparameters in practice, as we observe that GRAND is not very sensitive with those. Table 3 reports the best hyperparameters of GRAND we used for the results reported in Table 1.

Table 3: Hyperparameters of GRAND for results in Table 1.

| Hyperparameter | Cora | Citeseer | Pubmed |
|----------------|------|----------|--------|
| DropNode probability $\delta$ | 0.5 | 0.5 | 0.5 |
| Propagation step $K$ | 8 | 2 | 5 |
| Data augmentation times $S$ | 4 | 2 | 4 |
| CR loss coefficient $\lambda$ | 1.0 | 0.7 | 1.0 |
| Sharpening temperature $T$ | 0.5 | 0.3 | 0.2 |
| Learning rate | 0.01 | 0.01 | 0.2 |
| Early stopping patience | 200 | 200 | 100 |
| Hidden layer size | 32 | 32 | 32 |
| L2 weight decay rate | 5e-4 | 5e-4 | 5e-4 |
| Dropout rate in input layer | 0.5 | 0.0 | 0.6 |
| Dropout rate in hidden layer | 0.5 | 0.2 | 0.8 |

**Robustness Analysis in Section 4.5.** For random attack, we implement the attack method with Python and NumPy library. The propagation step $K$ of GRAND (with or without CR) is set to 5. And the other hyperparameters are set to the values in Table 3. As for Metattack [11], we use the publicly available implementation[3] published by the authors with the same hyperparameters used in

the original paper. We observe GRAND (with or without CR) is sensitive to the propagation step $K$ under different perturbation rates. Thus we search $K$ from {5,6,7,8} for each perturbation rate. The other hyperparameters are fixed to the values reported in Table 3.

**Other Experiments.** For the other results reported in Section 4.2 -4.6, the hyperparameters used in GRAND are set to the values reported in Table 3 with one or two changed for the corresponding analysis.

**Baseline Methods.** For the results of GCN or GAT reported in Section 4.5-4.6, the learning rate is set to 0.01, early stopping patience is 100, L2 weight decay rate is 5e-4, dropout rate is 0.5. The hidden layer size of GCN is 32. For GAT, the hidden layer consists 8 attention heads and each head consists 8 hidden units.

## B Theorem Proofs

### B.1 Proof for Theorem 1

*Proof.* The expectation of $\mathcal{L}_{con}$ is:

$$\frac{1}{2}\sum_{i=0}^{n-1}\mathbb{E}\left[(\tilde{z}_i^{(1)}-\tilde{z}_i^{(2)})^2\right] = \frac{1}{2}\sum_{i=0}^{n-1}\mathbb{E}\left[\left((\tilde{z}_i^{(1)}-z_i)-(\tilde{z}_i^{(2)}-z_i)\right)^2\right]. \tag{7}$$

Here $z_i = \text{sigmoid}(\overline{\mathbf{A}}_i\mathbf{X}\cdot\mathbf{W})$, $\tilde{z}_i = \text{sigmoid}(\overline{\mathbf{A}}_i\widetilde{\mathbf{X}}\cdot\mathbf{W})$. For the term of $\tilde{z}_i - z_i$, we can approximate it with its first-order Taylor expansion around $\overline{\mathbf{A}}_i\mathbf{X}\cdot\mathbf{W}$, i.e., $\tilde{z}_i - z_i \approx z_i(1-z_i)(\overline{\mathbf{A}}_i(\widetilde{\mathbf{X}}-\mathbf{X})\cdot\mathbf{W})$. Applying this rule to the above equation, we have:

$$\frac{1}{2}\sum_{i=0}^{n-1}\mathbb{E}\left[(\tilde{z}_i^{(1)}-\tilde{z}_i^{(2)})^2\right] \approx \frac{1}{2}\sum_{i=0}^{n-1}z_i^2(1-z_i)^2\mathbb{E}\left[(\overline{\mathbf{A}}_i(\widetilde{\mathbf{X}}^{(1)}-\widetilde{\mathbf{X}}^{(2)})\cdot\mathbf{W})^2\right]$$
$$= \sum_{i=0}^{n-1}z_i^2(1-z_i)^2\text{Var}_\epsilon\left(\overline{\mathbf{A}}_i\widetilde{\mathbf{X}}\cdot\mathbf{W}\right). \tag{8}$$

□

### B.2 Proof for Theorem 2

*Proof.* Expanding the logistic function, $\mathcal{L}_{org}$ is rewritten as:

$$\mathcal{L}_{org} = \sum_{i=0}^{m-1}\left[-y_i\overline{\mathbf{A}}_i\mathbf{X}\cdot\mathbf{W} + \mathcal{A}(\overline{\mathbf{A}}_i,\mathbf{X})\right], \tag{9}$$

where $\mathcal{A}(\overline{\mathbf{A}}_i,\mathbf{X}) = -\log\left(\frac{\exp(-\overline{\mathbf{A}}_i\mathbf{X}\cdot\mathbf{W})}{1+\exp(-\overline{\mathbf{A}}_i\mathbf{X}\cdot\mathbf{W})}\right)$. Then the expectation of perturbed classification loss can be rewritten as:

$$\mathbb{E}_\epsilon(\mathcal{L}_{sup}) = \mathcal{L}_{org} + \mathcal{R}(\mathbf{W}), \tag{10}$$

where $\mathcal{R}(\mathbf{W}) = \sum_{i=0}^{m-1}\mathbb{E}_\epsilon\left[\mathcal{A}(\overline{\mathbf{A}}_i,\widetilde{\mathbf{X}})-\mathcal{A}(\overline{\mathbf{A}}_i,\mathbf{X})\right]$. Here $\mathcal{R}(\mathbf{W})$ acts as a regularization term for $\mathbf{W}$. To demonstrate that, we can take a second-order Taylor expansion of $\mathcal{A}(\overline{\mathbf{A}}_i,\widetilde{\mathbf{X}})$ around $\overline{\mathbf{A}}_i\mathbf{X}\cdot\mathbf{W}$:

$$\mathbb{E}_\epsilon\left[\mathcal{A}(\overline{\mathbf{A}}_i,\widetilde{\mathbf{X}})-\mathcal{A}(\overline{\mathbf{A}}_i,\mathbf{X})\right] \approx \frac{1}{2}\mathcal{A}''(\overline{\mathbf{A}}_i,\mathbf{X})\text{Var}_\epsilon\left(\overline{\mathbf{A}}_i\widetilde{\mathbf{X}}\cdot\mathbf{W}\right). \tag{11}$$

Note that the first-order term $\mathbb{E}_\epsilon\left[\mathcal{A}'(\overline{\mathbf{A}}_i,\mathbf{X})(\widetilde{\mathbf{X}}-\mathbf{X})\right]$ vanishes since $\mathbb{E}_\epsilon(\widetilde{\mathbf{X}}) = \mathbf{X}$. We can easily check that $\mathcal{A}''(\overline{\mathbf{A}}_i,\mathbf{X}) = z_i(1-z_i)$. Applying this quadratic approximation to $\mathcal{R}(\mathbf{W})$, we get the quadratic approximation form of $\mathcal{R}(\mathbf{W})$:

$$\mathcal{R}(\mathbf{W}) \approx \mathcal{R}^q(\mathbf{W}) = \frac{1}{2}\sum_{i=0}^{m-1}z_i(1-z_i)\text{Var}_\epsilon(\overline{\mathbf{A}}_i\widetilde{\mathbf{X}}\cdot\mathbf{W}). \tag{12}$$

□

# C  Additional Experiments

## C.1  Results on Large Datasets

Table 4: Statistics of Large Datasets.

|  | Classes | Features | Nodes | Edges |
|---|---|---|---|---|
| Cora-Full | 67 | 8,710 | 18,703 | 62,421 |
| Coauthor CS | 15 | 6,805 | 18,333 | 81,894 |
| Coauthor Physics | 5 | 8,415 | 34,493 | 247,962 |
| Aminer CS | 18 | 100 | 593,486 | 6,217,004 |
| Amazon Computers | 10 | 767 | 13,381 | 245,778 |
| Amazon Photo | 8 | 745 | 7,487 | 119,043 |

We also evaluate our methods on six relatively large datasets, i.e., Cora-Full, Coauthor CS, Coauthor Physics, Amazon Computers, Amazon Photo and Aminer CS. The statistics of these datasets are given in Table 4. Cora-Full is proposed in [1]. Coauthor CS, Coauthor Physics, Amazon Computers and Amazon Photo are proposed in [7]. We download the processed versions of the five datasets here[4]. Aminer CS is extracted from the DBLP data downloaded from `https://www.aminer.cn/citation`. In Aminer CS, each node corresponds to a paper in computer science, and edges represent citation relations between papers. These papers are manually categorized into 18 topics based on their publication venues. We use averaged GLOVE-100 [6] word vector of paper abstract as the node feature vector. Our goal is to predict the corresponding topic of each paper based on feature matrix and citation graph structure.

Following the evaluation protocol used in [7], we run each model on 100 random train/validation/test splits and 20 random initializations for each split (with **2000** runs on each dataset in total). For each trial, we choose 20 samples for training, 30 samples for validation and the remaining samples for test. We ignore 3 classes with less than 50 nodes in Cora-Full dataset as done in [7]. The results are presented in Table 5. The results of GCN and GAT on the first five datasets are taken from [7]. We can observe that GRAND *significantly outperforms GCN and GAT on all these datasets.*

Table 5: Results on large datasets.

| Method | Cora Full | Coauthor CS | Coauthor Physics | Amazon Computer | Amazon Photo | Aminer CS |
|---|---|---|---|---|---|---|
| GCN | $62.2 \pm 0.6$ | $91.1 \pm 0.5$ | $92.8 \pm 1.0$ | $82.6 \pm 2.4$ | $91.2 \pm 1.2$ | $49.9 \pm 2.0$ |
| GAT | $51.9 \pm 1.5$ | $90.5 \pm 0.6$ | $92.5 \pm 0.9$ | $78.0 \pm 19.0$ | $85.7 \pm 20.3$ | $49.6 \pm 1.7$ |
| GRAND | $\mathbf{63.5 \pm 0.6}$ | $\mathbf{92.9 \pm 0.5}$ | $\mathbf{94.6 \pm 0.5}$ | $\mathbf{85.7 \pm 1.8}$ | $\mathbf{92.5 \pm 1.7}$ | $\mathbf{52.8 \pm 1.2}$ |

## C.2  Efficiency Analysis

The efficiency of GRAND is mainly influenced by two hyperparameters: the propagation step $K$ and augmentation times $S$. Figure 5 reports the average per-epoch training time and classification accuracy of GRAND on Cora under different values of $K$ and $S$ with #training epochs fixed to 1000. It also includes the results of the two-layer GCN and two-layer GAT with the same learning rate, #training epochs and hidden layer size as GRAND.

From Figure 5, we can see that when $K = 2, S = 1$, GRAND outperforms GCN and GAT in terms of both efficiency and effectiveness. In addition, we observe that increasing $K$ or $S$ can significantly improve the model's classification accuracy at the cost of its training efficiency. In practice, we can adjust the values of $K$ and $S$ to balance the trade-off between performance and efficiency.

## C.3  Parameter Sensitivity

We investigate the sensitivity of consistency regularization (CR) loss coefficient $\lambda$ and DropNode probability $\delta$ in GRAND and its variants on Cora. The results are shown in Figure 6. We observe that

(a) Per-epoch Training Time

(b) Classification Accuracy

Figure 5: Efficiency Analysis for GRAND.

(a) CR loss coefficient $\lambda$

(b) DropNode probability $\delta$

Figure 6: Parameter sensitivity of $\lambda$ and $\delta$ on Cora.

their performance increase when enlarging the value of $\lambda$. As for DropNode probability, GRAND, GRAND_GCN and GRAND_GAT reach their peak performance at $\delta = 0.5$. This is because the augmentations produced by random propagation in that case are more stochastic and thus make GRAND generalize better with the help of consistency regularization.

## C.4 DropNode vs Dropout

(a) Cora

(b) Citeseer

(c) Pubmed

Figure 7: GRAND vs. GRAND_dropout.

We compare GRAND and GRAND_dropout under different values of propagation step $K$. The results on Cora, Citeseer and Pubmed are illustrated in Figure 7. We observe GRAND always achieve better

performance than GRAND_dropout, suggesting *DropNode is much more suitable for graph data augmentation.*

Figure 8: Over-smoothing: GRAND vs. GRAND_GCN & GRAND_GAT on Cora.

### C.5 GRAND vs. GRAND_GCN & GRAND_GAT

As shown in Table 1, GRAND_GCN and GRAND_GAT get worse performances than GRAND, indicating GCN and GAT perform worse than MLP under the framework of GRAND. Here we conduct a series of experiments to analyze the underlying reasons. Specifically, we compare the MADGap values and accuracies GRAND, GRAND_GCN and GRAND_GAT under different values of propagation step $K$ with other parameters fixed. The results are shown in Figure 8. We find that the MADGap and classification accuracy of GRAND increase significantly when enlarging the value of $K$. However, both the metrics of GRAND_GCN and GRAND_GAT have little improvements or even decrease. This indicates that *GCN and GAT have higher over-smoothing risk than MLP.*

### C.6 Performance of GRAND under different label rates

We have conducted experiments to evaluate GRAND under different label rates. For each label rate setting, we randomly create 10 data splits, and run 10 trials with random initialization for each split. We compare GRAND with GCN and GAT. The results are shown in Table 6. We observe that GRAND consistently outperforms GCN and GAT across all label rates on three benchmarks.

Table 6: Classification Accuracy under different label rates (%).

| Dataset | Cora | | | Citeseer | | | Pubmed | | |
|---|---|---|---|---|---|---|---|---|---|
| Label Rate | 1% | 3% | 5% | 1% | 3% | 5% | 0.1% | 0.3% | 0.5% |
| GCN | 62.8±5.3 | 76.1±1.9 | 79.6±2.1 | 63.4±2.9 | 70.6±1.7 | 72.2±1.1 | 71.5±2.1 | 77.5±1.8 | 80.8±1.5 |
| GAT | 64.3±5.8 | 77.2±2.4 | 80.8±2.1 | 64.4±2.9 | 70.4±1.9 | 72.0±1.3 | 72.0±2.1 | 77.6±1.6 | 80.6±1.2 |
| GRAND | **69.1±4.0** | **79.5±2.2** | **83.0±1.6** | **65.3±3.3** | **72.3±1.8** | **73.8±0.9** | **74.7±3.4** | **81.4±2.1** | **83.8±1.3** |

## Footnotes

[1]`https://github.com/tkipf/pygcn`

[2]`https://pytorch-geometric.readthedocs.io`

[3]`https://github.com/danielzuegner/gnn-meta-attack`

[4]`https://github.com/shchur/gnn-benchmark`