[Reviews · NeurIPS 2020]

Review 1

Summary and Contributions: This paper proposes a random propagation strategy to perform graph-based data augmentation. Specifically, they randomly dropping out some input nodes, and perform graph propagation to get augmented features. The augmented features can then be used to train any neural networks, with a consistency regularization to enhance the prediction consistency of unlabeled nodes.

Strengths: (1) The framework is simple, yet it beats a wide range of GNN baselines on the semi-supervised node classification benchmark. (2) The proposed method is more robust to adversarial attacks. (3) The code is provided.

Weaknesses: The proposed methods are not that novel. Graph-based feature propagation is used in SGC (https://arxiv.org/abs/1902.07153) and GFN (https://arxiv.org/pdf/1905.04579.pdf); consistency regularization is a standard technique in semi-supervised learning (https://arxiv.org/abs/1610.02242). More specifically: (1) It seems that the consistency regularization is a general framework that can combine with other data augmentation methods, such as dropedge, and sampling algorithms. It would be better if the authors can also try these combinations, instead of only adopting their proposed dropnode augmentation. (2) The authors claim that the methods is suitable for semi-supervised node classification where training nodes are much scarcer. Thus, it would be better if the authors can provide a curve showing the performance of the proposed framework against other baselines under different training data percentage. (3) For larger datasets, it would be better if the authors can try on the Stanford OGB benchmark (https://ogb.stanford.edu/docs/leader_nodeprop/), which provides more standard data split. (4) For sampling-based baselines, better to add some recent works such as GraphSAINT (https://arxiv.org/abs/1907.04931) and LADIES (https://arxiv.org/abs/1911.07323). Also, better to combine these methods with some advanced base GNN. ========================================= Update: Overall, the rebuttal solves many of my concerns, so I decide to raise up the score. But I still have some comments on the added experiments: (1) The experiment added by the authors, which combines CR with Drop_Edge, and sees consistent performance enhancement against just Drop_Edge, further supports that CR is a general framework that can be used for many other sampling methods, which is very interesting and worth further researching. (2) Actually I'm more curious about the results with more data than the current split (say, 30% or more). But the current results added by the authors are very impressive, showing that GRAND is very powerful when label is extremely scarce. (3) The authors provide a theoretical analysis that the adding CR loss is approximate to concol a weighted average of the variance of node feature under different perturbation. What if we directly utilize this loss as regularization to train the model?

Correctness: Yes, they are correct.

Clarity: This paper is well written and motivated.

Relation to Prior Work: Yes.

Reproducibility: Yes

Additional Feedback:


Review 2

Summary and Contributions: This paper proposes a novel framework, GRAND, for semi-supervised learning on graphs. To improve the generalization ability and overcome the over-smoothing problem, the authors propose a random propagation strategy (DropNode) on graph data, i.e. randomly selecting and dropping the entire nodes. Also, a consistency regularization loss is proposed for unlabeled data to optimize the consistency among DropNode augmentations. Experiment results demonstrate the effectiveness of the proposed framework.

Strengths: The proposed framework is novel. The authors theoretically discuss the effectiveness of DropNode and the advantage towards Dropout. The empirical evaluation is sufficient. The ablation study demonstrates the proposed framework achieves better generalization ability and is more robust against attacks. I have carefully read the authors' rebuttal and other reviewers' comments, I still stand on my original rating.

Weaknesses: I think the overall framework is interesting and suitable for semi-supervised graph learning. However, the technical novelty of DropNode needs further discussion. With the proposed DropNode, a part of nodes is randomly selected and dropped, generating multiple perturbed graphs for training. This idea is similar to sub-graph sampling strategies proposed in FastGCN [7]. Another similar idea is seen in GraphSAGE [16], where neighbors are randomly sampled from the full neighborhood set of a node.

Correctness: As far as I’m concerned, the claims and methods in this paper are correct.

Clarity: This paper is well-written and easy to follow.

Relation to Prior Work: DropEdge [29] is a related work and the authors discuss the disadvantages of DropEdge from the aspect of computation complexity. However, the essential difference between DropEdge and DropNode is not discussed in the paper. I think the edge-orient method and node-orient method both can sample sub-graphs randomly and improve generalization ablity. It is necessary to explain why DropNode can bring significant performance gains for semi-supervised learning while DropEdge cannot.

Reproducibility: Yes

Additional Feedback:


Review 3

Summary and Contributions: This paper proposes a method for semi-supervised learning on graphs that address issues of over-smoothing and non-robustness. The method does so by applying graph data augmentation in combination with consistency regularization across nodes. The authors draw connections between their contributions and regularization and further show that the propose method, while simple, outperforms 14 state-of-the-art GNN baselines.

Strengths: * Clearly motivated method, accompanied by explanations of underlying insights * Theoretical backing, where the proposed method is rephrased as a regularization term * Thorough experiments and ablation studies

Weaknesses: No weaknesses to me.

Correctness: Yes and yes.

Clarity: The paper is well written.

Relation to Prior Work: Yes.

Reproducibility: Yes

Additional Feedback: I would have benefited from seeing the training splits in the main paper instead of the supplement, but regardless, the information was easy to find. I'm curious how the performance of GRAND responds as the number of training nodes is increased. I'd be interested to see what the limits are here - at how many labeled training nodes is performance comparable to existing baselines? Is there a lower limit of labeled training nodes at which this method is useful?

[Author Response · NeurIPS 2020]

<sup></sup>

**Reviewer #1:** Thank you for the positive comments and suggestions! Below we address your questions in detail.

*R1Q1.* **It would be better if authors can try dropedge and sampling methods, instead of only adopting dropnode.**

We have to emphasize that GRAND is a general framework, and random propagation can be achieved through various sampling methods, including DropNode, dropout, DropEdge or other methods. Indeed, besides DropNode, we also tried dropout as augmentation method in paper. In the past of few days, we have tried to implement another variant—GRAND_DropEdge, which adopts DropEdge as perturbation method. Specifically, we randomly remove some elements from adjacency matrix and use the perturbed matrix to perform mix-order propagation. Table 6 shows the classification results on benchmarks. we run 100 random trials for each dataset. As we can see, though GRAND_DropEdge gets worse performance than GRAND (DropNode) and GRAND_dropout, it still outperforms most of baselines (Cf. Table 1).

Table 6: Results of GRAND_DropEdge (%).

|  | Cora | Citeseer | Pubmed |
|---|---|---|---|
| GRAND_DropEdge | 84.5 ± 0.3 | 74.4 ± 0.4 | 80.9 ± 0.9 |

*R1Q2.* **It would be better if authors can provide the performance under different training ratio.**

We have conducted experiments to evaluate GRAND under different label rates. For each label rate setting, we randomly create 10 data splits, and run 10 trials with random initialization for each split. We compare GRAND with GCN and GAT. The results are shown in Table 7. We observe that GRAND consistently outperforms GCN and GAT across all label rates on three benchmarks.

Table 7: Classification Accuracy under different label rates (%).

| Dataset | Cora | | | | | Citeseer | | | | | Pubmed | | | | |
|---|---|---|---|---|---|---|---|---|---|---|---|---|---|---|---|
| Label Rate | 1% | 2% | 3% | 4% | 5% | 1% | 2% | 3% | 4% | 5% | 0.1% | 0.2% | 0.3% | 0.4% | 0.5% |
| GCN | 62.8±5.3 | 71.5±2.3 | 76.1±1.9 | 78.6±1.6 | 79.6±2.1 | 63.4±2.9 | 68.3±2.1 | 70.6±1.7 | 71.5±1.3 | 72.2±1.1 | 71.5±2.1 | 75.5±1.9 | 77.5±1.8 | 79.3±1.7 | 80.8±1.5 |
| GAT | 64.3±5.8 | 73.5±2.8 | 77.2±2.4 | 79.9±2.0 | 80.8±2.1 | 64.4±2.9 | 68.6±1.9 | 70.4±1.9 | 71.4±1.3 | 72.0±1.3 | 72.0±2.1 | 75.7±2.0 | 77.6±1.6 | 79.3±1.7 | 80.6±1.2 |
| GRAND | 69.1±4.0 | 76.7±2.5 | 79.5±2.2 | 82.7±1.6 | 83.0±1.6 | 65.3±3.3 | 70.0±2.0 | 72.3±1.8 | 73.0±1.1 | 73.8±0.9 | 74.7±3.4 | 79.9±2.5 | 81.4±2.1 | 82.7±1.6 | 83.8±1.3 |

*R1Q3.* **The proposed methods are not that novel.**

Though some related techniques have been studied in previous works, our work is not a trivial combination of them. Our contributions lie in the general framework for graph-based semi-supervised learning. The proposed method is simple yet effective and beats 14 state-of-the-art baselines on standard benchmarks. More importantly, the framework has a well-established theoretical guarantee. To the best of our knowledge, this is the first work that provides theoretical explanations for applying consistency regularization on graph data.

*R1Q4.* **It would be better if the authors can try Stanford OGB ... add GraphSAINT & LADIES**

We've reported the results of 14 GNN baselines on widely-used datasets—Cora, Citeseer, and Pubmed, as well as the performance of 6 more datasets in Appendix. Due to the limited response time, we prioritize experiments for Q1 and Q2. We will have results on OGB and discuss and compare LADIES and GraphSAINT in next version.

**Reviewer #2:** Thank you for the positive comments! We address your specific concerns in detail below.

*R2Q1.* **The technical novelty of DropNode needs further discussion, which is similar to FastGCN/GraphSAGE.**

Since GRAND is a general framework, the sampling methods used in FastGCN/GraphSAGE can also be used in GRAND for graph data augmentation. Here we'd like to talk more about the differences between these two kinds of sampling methods. In terms of the objective, FastGCN/GraphSAGE is mainly used for scaling and accelerating GCNs, while DropNode is aimed to perform graph data augmentation to improve model's generalization capacity. Technically, GraphSAGE adopts node-wise sampling, which is much less efficient than DropNode as it requires recursive sampling of neighborhoods for every node. FastGCN uses layer-wise importance sampling, in which the sampling probability is related to node's degree. Compared to that, DropNode samples each node based on an i.i.d. Bernoulli distribution, which will generate more stochastic augmentations and thus works better than importance sampling in improving model's generalization. Overall, the sampling methods used in FastGCN/GraphSAGE are not optimal options for graph data augmentation when compared with DropNode. One promising research direction is to combine these two kinds of sampling methods to make GRAND more scalable. We will leave this part into our future work.

*R2Q2.* **Why DropNode can bring significant performance gains while DropEdge cannot?**

As we claimed in paper (line 235 & line 229), the results of DropEdge reported in Table 1 are taken from its original paper [29], wherein GCN serves as the backbone model. To make narration more clear, here we rename it GCN_DropEdge. GCN_DropEdge cannot get as much performance improvement as GRAND for two reasons. First, GCN_DropEdge does not use consistency regularization, the critical strategy for improving model's generalization used in GRAND. Second, in GCN_DropEdge, feature propagation is coupled with nonlinear transformation, leading to oversmoothing and overfitting issues when increasing model's layer num. This causes GCN_DropEdge cannot perform high-order message passing like GRAND. In recent days, we have incorporated DropEdge into GRAND framework, named GRAND_DropEdge (Cf. the response to *R1Q1*). GRAND_DropEdge does not have the above limitations and outperforms GCN_DropEdge by a large margin (Cf. Table 6).

**Reviewer #4:** Thank you for the encouraging comments! We have provided the classification results under different label rates in Table 7. We will provide more analysis on the limitations of GRAND in the next version.

[Meta-Review · NeurIPS 2020]

All reviewers appreciate the idea of the paper, its simplicity and its good empirical performance.